# Oxidative Mechanisms and Cardiovascular Abnormalities of Cirrhosis and Portal Hypertension

**DOI:** 10.3390/ijms242316805

**Published:** 2023-11-27

**Authors:** Hongqun Liu, Henry H. Nguyen, Sang Youn Hwang, Samuel S. Lee

**Affiliations:** Liver Unit, University of Calgary Cumming School of Medicine, Calgary, AB T2N 4N1, Canadahhnguye@ucalgary.ca (H.H.N.); mongmani@daum.net (S.Y.H.)

**Keywords:** oxidative stress, cardiovascular, cirrhosis, liver, portal hypertension

## Abstract

In patients with portal hypertension, there are many complications including cardiovascular abnormalities, hepatorenal syndrome, ascites, variceal bleeding, and hepatic encephalopathy. The underlying mechanisms are not yet completely clarified. It is well known that portal hypertension causes mesenteric congestion which produces reactive oxygen species (ROS). ROS has been associated with intestinal mucosal injury, increased intestinal permeability, enhanced gut bacterial overgrowth, and translocation; all these changes result in increased endotoxin and inflammation. Portal hypertension also results in the development of collateral circulation and reduces liver mass resulting in an overall increase in endotoxin/bacteria bypassing detoxication and immune clearance in the liver. Endotoxemia can in turn aggravate oxidative stress and inflammation, leading to a cycle of gut barrier dysfunction → endotoxemia → organ injury. The phenotype of cardiovascular abnormalities includes hyperdynamic circulation and cirrhotic cardiomyopathy. Oxidative stress is often accompanied by inflammation; thus, blocking oxidative stress can minimize the systemic inflammatory response and alleviate the severity of cardiovascular diseases. The present review aims to elucidate the role of oxidative stress in cirrhosis-associated cardiovascular abnormalities and discusses possible therapeutic effects of antioxidants on cardiovascular complications of cirrhosis including hyperdynamic circulation, cirrhotic cardiomyopathy, and hepatorenal syndrome.

## 1. Introduction

Oxidative stress is an imbalance between the production of reactive oxygen species (ROS) and the antioxidant system reducing its capability to detoxify ROS or repair the resulting damage, i.e., ROS overwhelms antioxidants. Oxidative stress is a key pathogenic factor in chronic liver injury of various etiologies, such as alcoholic liver disease [1], nonalcoholic fatty liver diseases (NAFLD) [2], chronic viral hepatitis, and cholestatic diseases. This damage will ultimately lead to cirrhosis (defined as hepatic architectural damage characterized by nodular regeneration and diffuse fibrosis) [3]. Furthermore, oxidative stress also plays an important pathogenic role in portal hypertension (defined as a portal venous pressure of greater than 12 mm Hg) [4], cirrhotic cardiomyopathy (CCM) [5], hepatorenal syndrome (HRS) [6], cirrhosis-related pulmonary complications [7], and hepatic encephalopathy [8]. The present review aims to summarize the role of oxidative stress in cirrhosis-related cardiovascular changes including hyperdynamic circulation, CCM, cardiac arrhythmias, acute kidney injury (AKI), and hepatorenal syndrome (HRS). We will also review a potential therapeutic role of antioxidants in cardiovascular anomalies of cirrhosis.

## 2. Pathogenic Mechanisms of Oxidative Stress

Oxidative stress plays a crucial role in the progression of chronic liver diseases to cirrhosis and the development of associated complications. The liver plays a central role in the detoxification of endogenous and exogenous toxins, and harbors a high antioxidant function [3]. In cirrhosis, liver function is significantly impaired and therefore antioxidant function is also jeopardized. This is coupled with excess oxidative stress resulting from portal hypertension driven by gastrointestinal congestion and bacterial translocation [9]. The overactivated ROS can damage the intestine structurally and functionally, as increased lipid peroxidation and protein oxidation have been found in the intestinal mucosa of cirrhotic rats [10] and decompensated cirrhotic patients [11]. Using a carbon tetrachloride (CCl_4_) model of cirrhosis, Ramachandran and coworkers [10] evaluated oxidative stress status in the gastrointestinal tract. In comparison with controls, xanthine oxidase (XO) activity (an index of oxidative stress) was significantly increased, and xanthine dehydrogenase activity (a parameter of antioxidant status) was significantly decreased in the intestine of cirrhotic rats. This alteration of oxidative stress was associated with a significant reduction in villus fraction, increased enterocyte necrosis, loss of tight junctions, and abnormal intestinal brush border. The damaged intestinal mucosa is thought to enhance intestinal permeability and bacterial translocation that in combination with collateral circulation and an impaired liver results in endotoxemia. Endotoxemia can then further increase oxidative stress, intestinal barrier dysfunction, and organ injury, leading to a ‘vicious cycle’: gut barrier dysfunction → endotoxemia → organ injury. Endotoxemia in cirrhotic subjects plays critical role in cirrhotic complications such as CCM [11], and AKI/HRS [12].

## 3. Overview of Cardiovascular Abnormalities of Cirrhosis and Portal Hypertension

The cardiovascular system in cirrhotic patients is abnormal, and is characterized by portal hypertension, systemic hyperdynamic circulation (increased cardiac output and decreased peripheral vascular resistance), and arterial pressure [13]. The decrease in peripheral vascular resistance is thought to be due to an imbalance between vasoconstrictive and vasodilatory factors. The former is mainly driven by the sympathetic system [14], renin–angiotensin system (RAS) [15], endothelin-1 [16], and thromboxane [17]; the latter by glucagon [18], prostaglandins [19], bile acids, nitric oxide (NO) [20], carbon monoxide (CO) [21], and hydrogen sulfate (H_2_S) [22]. Although the vasoconstrictors such as the sympathetic system [23,24] and RAS [25,26] are increased in subjects with cirrhosis, the response of cirrhotic subjects to vasoconstrictors as a whole is impaired [27]. Vasodilators play a dominant role in cirrhotic patients and therefore the cardiovascular system in patients with cirrhosis is characterized by peripheral vascular dilation, decreased systemic vascular resistance (SVR), and mean arterial pressure (MAP). The cardiac output is increased at rest with cirrhotic patients having been described to have hyperdynamic circulation [22]. Under stress, the cardiac functional reserve is insufficient in cirrhosis, with decreased left ventricular responsiveness having been described [28]. Moreover, cardiac diastolic dysfunction has also been reported and can lead to significantly smaller increases in stroke volume when challenged [28]. This collection of cardiac functional abnormalities in patients with cirrhosis is called cirrhotic cardiomyopathy. The diagnosis of cirrhotic cardiomyopathy is based on advanced imaging examination at rest [29].

The pathophysiology of cardiovascular changes in cirrhosis is multifaceted, with inflammation (evidenced by increases of proinflammatory cytokines) [30] and oxidative stress as significant contributors to these changes [5]. There is evidence that oxidative stress also plays an essential role in non-cirrhotic cardiovascular diseases [31]. However, the role of oxidative stress in the pathophysiology of cardiovascular changes in cirrhosis remains incompletely clarified.

## 4. Oxidative Stress in Pathogenesis of Hyperdynamic Circulation

Cirrhosis and portal hypertension cause hyperdynamic circulation. There are two theories explaining hyperdynamic circulation in subjects with cirrhosis and portal hypertension: the humoral theory and central neural dysregulation [9].

## 5. Oxidative Stress in the Humoral Theory

The humoral hypothesis suggests that in cirrhosis, mesenteric congestion causes endotoxemia. Endotoxin stimulates the generation of vasodilators such as glucagon [18], prostaglandins [19], NO [32], CO [33], and H_2_S [34]. All these vasodilators additively/synergically dilate peripheral vasculature which results in hyperdynamic circulation. 

Oxidative stress also plays an important role in hyperdynamic circulation. In bile duct ligation (BDL)-induced cirrhosis in rats, Lee et al. [35] found that mesenteric markers of oxidative stress, such as thiobarbituric acid reactive substances (TBARS, an index of lipoperoxidation), and malondialdehyde (MDA), are significantly increased. Congruent with these data, our group has also shown that mesenteric myeloperoxidase (MPO) is significantly increased in portal hypertensive rats [9]. Furthermore, the levels of oxidative stress in the mesentery are closely related to circulating proinflammatory cytokines, such as TNF-α, IL-1β, and IL-6. Lee’s study suggests that oxidative stress may have an additive/synergic effect on systemic inflammation on hyperdynamic circulation in cirrhotic animal models [35]. Now it is clear that cellular enzymes called nicotinamide adenine dinucleotide phosphate (NADPH) oxidases produce a considerable amount of ROS in humans [36] and a rat model of partial portal vein ligation (PPVL) [37]. Deng and coworkers [37] demonstrated that H_2_O_2_ is significantly increased in mesenteric tissues and in parallel, phosphorylated eNOS (p-eNOS) is elevated. NADPH oxidase inhibitor, GKT137831, significantly reduces mesenteric H_2_O_2_ and p-eNOS, linking oxidative stress and the vasodilator, NO [37]. Interestingly, GKT137831 also reduces cardiac index, portal vein pressure, portal vein blood flow, and portal–systemic shunting in PPVL rats. GKT137831 reverses the decreased mesenteric artery contractile response to norepinephrine in PPVL rats. The final effect of NADPH oxidase inhibition is ameliorating hyperdynamic circulation [37].

In 1998, the Moore lab [38] demonstrated that PPVL rats develop hyperdynamic circulation; this was reversed when these rats were treated with N-acetylcysteine (NAC) (Figure 1). Licks et al. [39] later found that in PPVL rats, in parallel with hyperdynamic circulation, the levels of gastric TBARS, nitrates, and nitrites are increased. In their four groups of rats (Sham, Sham + NAC, PPVL, and PPVL + NAC), the levels of oxidative stress markers paralleled that of nitrates and nitrites. The antioxidant parameters, such as superoxide dismutase (SOD) and glutathione peroxidase (GPx), were significantly decreased in PPVL rats. Histology showed that the vessels in the gastric mucosa in the PPVL group were dilated. NAC treatment significantly reversed these changes and resulted in a circulation that resembled what was observed in sham operated control rats. With these results, they concluded that oxidative stress contributes to portal hypertension and hyperdynamic circulation via the regulation of nitrates and nitrites, and antioxidant reverses these changes in the rat PPVL model.

Interestingly, Iwakiri et al. [40] used a double knockout (iNOS, eNOS) mice in a PPVL model and observed that these mice still develop hyperdynamic circulation. This suggests that other variables in addition to the humoral theory may be at play in this complex condition. 

## 6. Oxidative Stress in Central Neural Dysregulation 

Hyperdynamic circulation due to the dysregulation of the central neural system [41] has also been described in subjects with cirrhosis and portal hypertension. This theory proposes that there is a reflex arc which regulates the cardiovascular system. The reflex arc includes the receptors in mesentery, afferent nerves, cardiovascular nuclei, and efferent nerves. The signals originating from receptors in mesentery are relayed via afferent nerves to the central neural system which then dispatches signals to the cardiovascular system via efferent nerves. The integrity of the reflex arc is essential for the regulation of the cardiovascular system [9,41,42]. Portal hypertension causes congestion in the mesentery which activates chemoreceptors and/or baroreceptors in the splanchnic area. Our study found that 10 days after PPVL, the MAP and SVR are significantly decreased and cardiac output significantly increased, indicating a hyperdynamic circulation [42]. We examined different interventions to test the role of this reflex arc in hyperdynamic circulation. We first tested the role of capsaicin-sensitive nerves in hyperdynamic circulation. Capsaicin was used to denervate the afferent nerves in rat pups. These rats were subjected to PPVL or BDL-induced cirrhosis when they reached adulthood [43]. We showed that capsaicin-treated PPVL or cirrhotic rats had similar cardiac output and systemic vascular resistance compared to the sham-operated group. In comparison, the PPVL or cirrhotic rats treated with vehicle (DMSO + ethanol) demonstrated hyperdynamic circulation. These data confirmed that capsaicin-treated rats have no capacity to develop hyperdynamic circulation when subjected to PPVL or cirrhosis. Capsaicin had no hemodynamic effect on sham-operated rats. These results showed that capsaicin treatment blocks the development of vasodilation in cirrhotic and portal hypertensive rats. Thus, primary afferent innervation is important in the pathogenesis of hyperdynamic circulation in portal hypertension and cirrhosis. 

We then tested the central neural system in the regulation of hyperdynamic circulation in PPVL rats [42]. *c-fos* is an immediate-early gene and has important roles in cellular signal transduction. The *c-fos* protein product Fos is significantly increased in central cardiovascular nuclei such as the nucleus tractus solitarius (NTS) and paraventricular nucleus (PVN). Fos is an activation marker in the central neural system. We found that Fos expression is the prerequisite of hyperdynamic circulation in PPVL rats. Fos is detectable at day 1 after PPVL with immunohistochemistry and persistently increased when examined daily in the PPVL rats. However, the hyperdynamic circulation developed on day 3 and remained thereafter. When Fos expression in the NTS was blocked by local microinjection of *c-fos* antisense oligonucleotides, the increased cardiac output decreased SVR and MAP were reversed in PPVL rats. *c-fos* antisense oligonucleotides had no effect on circulation in sham-control rats. This experiment indicated that cardiovascular nuclei are crucial to the regulation of hyperdynamic circulation in PPVL rats [42]. 

We then tried to clarify the afferent nervous pathway. An inflatable cuff around the portal vein was used to acutely increase the portal pressure, then the vagal nerve electrical activity in the cervical area was recorded [41]. The acute increase in portal pressure immediately increased the vagal nerve electrical activity. In another experiment, the cervical vagus nerve was ablated by capsaicin. Three weeks after vagal ablation, the rats were subjected to PPVL. Our results showed that vagal nerve ablation significantly decreased Fos expression in the PVN of PPVL rats. Furthermore, vagal nerve blocked the development of hyperdynamic circulation in PPVL rats. This implied that an intact vagal nerve is a sine qua non in the pathogenesis of hyperdynamic circulation in PPVL rats [41]. 

The initial signal that triggers the hyperdynamic circulation in PPVL rats is thought to be mesenteric congestion. The congestion impacts two types of receptors, baroreceptors and chemoreceptors, as congestion not only increases the mesenteric venous pressure, but also causes ischemia. Chen et al. [44] induced cirrhosis in rats by CCl_4_, and reported that H_2_O_2_ content in the mesenteric arterial wall was significantly increased in comparison to control animals. Several studies demonstrated that an antioxidant such as NAC significantly reverses hyperdynamic circulation in portal hypertensive animal models. 

One of these studies from the Moore Lab demonstrated that NAC alleviates oxidative stress and prevents hyperdynamic circulation in the PPVL rats. They showed that the portal pressure remains significantly higher in the PPVL + NAC animals compared with sham + NAC rats [38]. However, that study did not further investigate the mechanism of NAC on the attenuation of hyperdynamic circulation. 

We therefore performed a study to investigate the role of oxidative stress in the pathogenesis of hyperdynamic circulation, specifically aiming to evaluate whether oxidative stress is the initiating signal in the gut. We also used the PPVL model in rats because prehepatic portal hypertension creates mesenteric congestion and ischemia without significant liver parenchymal injury. Our study also reconfirmed the effect of NAC on hyperdynamic circulation in PPVL rats [10] (Figure 2). Furthermore, we found that jejunal MPO, an index of intestinal oxidative stress, is significantly increased in the PPVL model in rats; NAC treatment significantly decreased the activity of jejunal MPO (Figure 3). Interestingly, NAC also significantly decreased cardiac output, increased MAP and SVR, and reversed the hyperdynamic circulation in PPVL rats (Figure 2). To confirm that it is the oxidative stress that inaugurates hyperdynamic circulation, H_2_O_2_ was applied directly to the mesenteric area. We confirmed that H_2_O_2_ stimulates Fos expression in PVN (Figure 4), and decreased MAP, a direct index of hyperdynamic circulation. All these data indicate that it is the oxidative stress that triggers the hyperdynamic circulation in portal hypertension [9].

## 7. Cirrhotic Cardiomyopathy

Cardiac dysfunction underlying cirrhosis in the absence of pre-existing heart disease is known as CCM [29]. The diagnostic criteria of CCM include systolic and diastolic dysfunction (Table 1). The mechanisms of CCM include endotoxemia, increased proinflammatory cytokines such as TNFα and IL-1β [45], apoptosis [46], changes in cardiac myofilament proteins [47], and oxidative stress [48]. 

ROS were previously thought to be produced almost exclusively from mitochondrial metabolism. Now it is clear that cellular enzymes called NADPH oxidases produce a considerable amount of ROS in humans [36] and animal models of PPVL in rats [37]. NADPH is located predominantly in the cytosolic compartment while NADH is localized predominantly to mitochondria and therefore, mitochondria are a main site of ROS production [49]. The first three cellular sources of ROS are neutrophils, monocytes, and cardiomyocytes. ROS is overproduced in subjects with cirrhosis [50]. 

It is well known that the heart is an energy-consuming organ and around one-third of the cardiomyocyte is mitochondria in adults [49]. Ninety-five percent of energy supplied to the cardiomyocyte is derived from mitochondria (ref) [51]. Our study demonstrated that monocytes are increased in the hearts of cirrhotic animals [52] and highlights the innate immune response as a contributing source of ROS. Mousavi and colleagues [5] evaluated oxidative stress in cirrhotic hearts induced by BDL in rats and found that the ROS level, lipid peroxidation, and protein carbonylation were significantly increased in cirrhotic hearts compared with those in control hearts. Glutathione (GSH) in the cirrhotic heart was significantly depleted and oxidized glutathione (GSSG) was significantly increased, and the GSH/GSSG ratio was significantly decreased. The total antioxidant capacity was significantly reduced. Moreover, Mousavi’s study revealed that the content of ATP, a marker of myocardial mitochondrial function, was significantly decreased (less than 50% of controls) in cirrhotic animals [5]. When mitochondria cannot meet the demands of a cell for ATP, ROS will be produced [49]. Mitochondria generate many types of ROS, including superoxide anion (O_2_^−^), H_2_O_2_, and hydroxyl radical (HO). Although the reactivity is different in each individual ROS, they all cause cardiomyocyte dysfunction. Mitochondrial homeostasis is therefore vital for preserving cardiac function [53]. The dysfunction of mitochondria causes a cellular energy crisis which plays an essential role in CCM (Figure 5). Our previous study also demonstrated that in the cirrhotic rat heart, 2,4-dinitrophenylhydazone, an indicator of oxidative stress, is significantly increased, and Nrf2, an antioxidant protein, is significantly decreased. Erythropoietin, an antioxidant, significantly reduced oxidative stress and augmented antioxidant proteins. Furthermore, erythropoietin significantly improved cardiac function in a rat cirrhotic model. These data imply that oxidative stress plays an important pathogenic role in the cirrhotic heart [48].

## 8. Role of Oxidative Mechanisms in Cardiac Arrhythmias

In patients with cirrhosis, atrial fibrillation is the most common arrhythmia [54]. Oxidative stress is significantly increased and antioxidants are significantly decreased in noncirrhotic patients with atrial fibrillation [55,56]. Corradi and colleagues [55] reported that molecular markers of oxidative stress such as heme oxygenase-1 and 3-nitrotyrosine were significantly elevated and antioxidant biomarkers such as SOD-2 were significantly reduced in patients with atrial fibrillation. Bezna et al. [56] studied 80 patients with supraventricular cardiac arrhythmias and 40 healthy volunteers, and reported that specific antioxidant biomarkers such as SOD and glutathione peroxidase were significantly decreased in all patients with arrhythmias. Is there a direct *causal effect* of increased oxidative stress in cardiac arrhythmias? Morita et al. [57] directly exposed isolated hearts to H_2_O_2_ (0.1 mM) in a Langendorff setup and demonstrated that H_2_O_2_ directly induces ventricular fibrillation. Ranolazine, which inhibits late inward sodium current and has an antioxidant effect, prevents/terminates H_2_O_2_-induced ventricular fibrillation. Morita’s study provides solid evidence that oxidative stress causes arrhythmias.

## 9. Acute Kidney Injury (AKI) and Hepatorenal Syndrome 

HRS is defined as renal failure developing in patients with pre-existing chronic liver failure (acute or chronic) in the absence of any other identifiable cause of renal disease [58].

HRS is a serious and life-threatening complication of decompensated cirrhosis. HRS is not a purely ‘functional’ renal failure due to hemodynamic perturbation, as oxidative stress and inflammation are also thought to play a significant role in the pathogenesis of this condition. Local/systemic oxidative stress and inflammation cause structural changes [59]. The International Club of Ascites in 2007 classified HRS as types 1 and 2 (HRS-1 and HRS-2) [60]. HRS-1 is defined as a rapid deterioration of renal function due to precipitating factors, such as bacterial infection, large volume paracentesis, and gastrointestinal hemorrhage [61]. HRS-2 is a relatively slow process of progressive renal dysfunction. HRS-2 usually has no obvious precipitating factors. HRS-1 therefore manifests as acute renal failure and HRS-2 is mainly characterized by refractory ascites. 

The pathogenesis of HRS is the underfilling of the arterial circulation due to arterial vasodilation combined with inadequate renal perfusion due to the ventricular dysfunction of CCM [62]. The causal connection between oxidative stress and HRS/AKI is whether oxidative stress contributes to arterial vasodilation and CCM. The pathogenic role of oxidative stress in CCM has been reviewed above. As for arterial vasodilatation, accumulating evidence indicates that oxidative stress contributes to the structural and functional derangement of the intestinal mucosa which results in the disruption of gut barrier integrity and increases permeability. The increased permeability, mesenteric congestion due to portal hypertension, and decreased hepatic detoxication capacity due to cirrhosis result in bacterial overgrowth, translocation, and increased lipopolysaccharides (LPS) which escape the liver via collateral circulation; the final result is endotoxemia. Endotoxin causes kidney damage in multifaceted ways [63] including vasodilation via TNFα, nitric oxide, and carbon monoxide. Therefore, oxidative stress is an Important factor in the pathogenesis of HRS. 

## 10. Antioxidants as Potential Treatment Options in Cardiovascular Anomalies of Cirrhosis

There is no accepted specific treatment for the management of cardiovascular anomalies of cirrhosis. Traditional therapeutic strategies for non-cirrhotic heart diseases, such as vasodilators, are not suitable for heart dysfunction in cirrhosis because cirrhotic patients often have vasodilation and hypotension. As such, vasodilators may worsen a cirrhotic patient’s clinical status [64]. Therefore, angiotensin-converting enzyme (ACE) inhibitors or angiotensin receptor blockers are not applicable in advanced cirrhosis and are contraindicated in HRS [64]. Liver transplantation is the definitive ‘cure’ for cardiovascular anomalies of cirrhosis. However, the shortage of donor organs may limit its application, and transplantation is not widely available in all global regions. Moreover, it is an expensive, complex procedure which is not feasible for many medical centers. Finally, long term immunosuppression is associated with adverse effects, including risk of infections and malignancy. Thus, the search for medical therapies for the cardiovascular complications of cirrhosis must continue. Accordingly, any study that finds a link between cardiovascular anomalies and certain molecules may lead to a potential therapeutic strategy. Oxidative stress-related molecules may thus be targets for this purpose.

## 11. Nonspecific Beta-Adrenergic Blockers (NSBBs) 

Taprantzia et al. [65] investigated the status of oxidative stress in cirrhotic patients and demonstrated that oxidative indicators such as lipid hydroperoxides and malondialdehyde were significantly increased in cirrhotic patients compared with healthy controls. Propranolol treatment for one month significantly reduced oxidative stress by decreasing portal pressure in cirrhotic patients [66], improving mesenteric venous congestion, and decreasing intestinal permeability [67]. These effects indirectly alleviated endotoxemia and systemic inflammation [65]. 

Improvement in systemic inflammation benefits the cardiovascular system. Another potential benefit is that NSBBs shorten the prolonged QTc interval and decrease the risk of ventricular arrhythmias [68]. However, Silvestre et al. [69], in a randomized controlled trial, treated CCM patients with metoprolol and did not find significant improvement of cardiac function after 6 months in the metoprolol-treated group compared to a placebo. A lack of significant response was thought to reflect heterogeneity of the patient population and sympathetic neural response relative to severity of cirrhosis [70]. 

## 12. Taurine 

Taurine has pleiotropic functions including anti-oxidation, anti-inflammation, and anti-apoptosis. It impacts many organs including retina, skeletal muscle, liver, platelets, and leukocytes [71]. Taurine is also thought to be essential for cardiovascular function as studies evaluating taurine transporter knockout mice note cardiac dysfunction as a phenotype [72]. Taurine has been shown to have a protective effect on oxidative stress-induced vascular dysfunction [73]. The role of taurine in CCM needs to be further investigated and characterized. The biosynthesis of taurine occurs primarily in the liver [71]. Cirrhosis decreases the functional liver mass and consequently the synthesis of taurine [64]. Low taurine serum levels have been closely associated with many oxidative stress-mediated pathologies, including hepatic disorders and cardiomyopathy [74]. 

Given that the antioxidant capacity in patients with cirrhosis is decreased, supplementation of taurine may be potentially beneficial. Taurine has been shown to reduce lipid peroxidation and protects cells from damage [75]. Using a model of transverse aortic constriction-induced heart failure in mice, Liu et al. [76] demonstrated that taurine has a protective effect on cardiac function. The mechanisms are thought to be secondary to reducing myocyte oxidative stress, apoptosis, hypertrophy, and cardiac fibrosis. These protective effects of taurine on non-cirrhotic heart failure may also apply to CCM. Mousavi and colleagues [5] showed that taurine significantly reduced tissue oxidative stress which includes lipid peroxidation, ROS, protein carbonylation, and the GSH/GSSG ratio in a bile duct ligation model of cirrhosis. Overall, taurine increased total antioxidant capacity and mitochondrial ATP content in this study. In summary, taurine decreases oxidative stress and improves mitochondrial function in the cirrhotic rat heart. Furthermore, taurine also decreases the level of creatine kinase MB (CK-MB), a marker of heart injury. Taurine is a valuable candidate worth further investigation in cirrhotic patients with cardiovascular complications.

## 13. Spermidine

Similar to taurine, spermidine also possesses antioxidant, anti-inflammatory, and anti-apoptotic properties [77,78]. Omar et al. [79] evaluated the effects of spermidine on isoproterenol-induced acute myocardial infarction and reported that it significantly increased electrocardiographic RR and QRS intervals to normal values, and decreased QT intervals and ST segment height to normal ranges, compared to the untreated group. Spermidine also significantly reduced serum CK-MB and lactate dehydrogenase (LDH), both parameters of serum cardiac injury. In parallel, the reduced antioxidant capacity in the untreated AMI group was rescued with spermidine treatment, indicating that the protective effect of spermidine is at least partially mediated via inhibition of oxidative stress [79]. 

Furthermore, Sheibani et al. [78] investigated the effects of spermidine in BDL-cirrhotic rats and demonstrated that spermidine significantly reduced the cardiac oxidative stress and inflammation. Moreover, spermidine significantly decreased the QTc interval in the BDL group (204 vs. 170 ms, *p* < 0.001). The cardiac contractility of spermidine-treated cirrhotic rats was also significantly increased in comparison with that from untreated BDL rats. These studies raise the possibility of the clinical application of spermidine in cirrhotic patients with cardiovascular diseases.

## 14. Direct Antioxidants

A meta-analysis [80] demonstrated that NAC not only significantly decreased oxidative markers, such as MDA and homocysteine, but also inflammatory markers, such as TNF-α and IL-6. Subjects with cirrhosis have both oxidative stress and inflammation, and this raises the question whether agents such as NAC can be used in this clinical setting. Using the BDL model of cirrhosis in rats, Lee and coworkers [35] reported that TBARS and MDA markers of oxidative stress were significantly increased in BDL vs. control rats. In parallel with this, inflammatory markers such as TNF-α, IL-1β, and IL-6 were also significantly increased in BDL rats. NAC significantly decreased both oxidative and inflammatory markers. Interestingly, one month treatment of NAC significantly attenuated systemic and splanchnic hyperdynamic circulation, improved hepatic endothelial dysfunction, and reduced intrahepatic resistance [27].

Another direct antioxidant is hydrogen. Because of the small size of the hydrogen molecule, it easily penetrates the cell membrane to the cytosol. Another advantage is that there are no side effects because it can be metabolized without residue [81]. Hydrogen exerts antioxidant [82], anti-inflammatory, and antiapoptotic effects [83,84], which has cardioprotective benefits. Hydrogen might be a novel treatment in various cardiovascular conditions such as ischemia–reperfusion injury [85] and cardiac transplantation because hydrogen protects cardiac allografts and reduces intimal hyperplasia of aortic allografts [86]. Lee et al. [35] revealed that hydrogen-rich saline significantly decreased TBARS and MDA, markers of oxidative stress, and increased SOD, GSH, markers of antioxidant in BDL rats compared with BDL + vehicle. Furthermore, hydrogen-rich saline decreased inflammatory markers such as TNF-α, IL-1β, and IL-6. The final results were the improvement of hepatic endothelial function, intrahepatic resistance, and systemic and splanchnic hyperdynamic circulation [35]. The studies on hydrogen at present are limited to animals, but there is potential for clinical application.

Other antioxidants, such as resveratrol [87], melatonin [88], and albumin [89] also have protective/therapeutic effects on cardiovascular diseases; these antioxidants may thus be applicable to treat cardiovascular complications in cirrhosis.

In conclusion, in subjects with cirrhosis and portal hypertension, oxidative stress plays an essential role in the pathogenesis of several complications. These complications include hyperdynamic circulation, cirrhotic cardiomyopathy, and acute kidney injury/hepatorenal syndrome. Oxidative stress stimulates inflammation and impacts the production of cardiac energy, which result in cardiac and vascular dysfunction. Antioxidants can reverse or mitigate these processes and thus may have potential therapeutic effects on cardiovascular and renal abnormalities in cirrhosis.

## Figures and Tables

**Figure 1 ijms-24-16805-f001:**
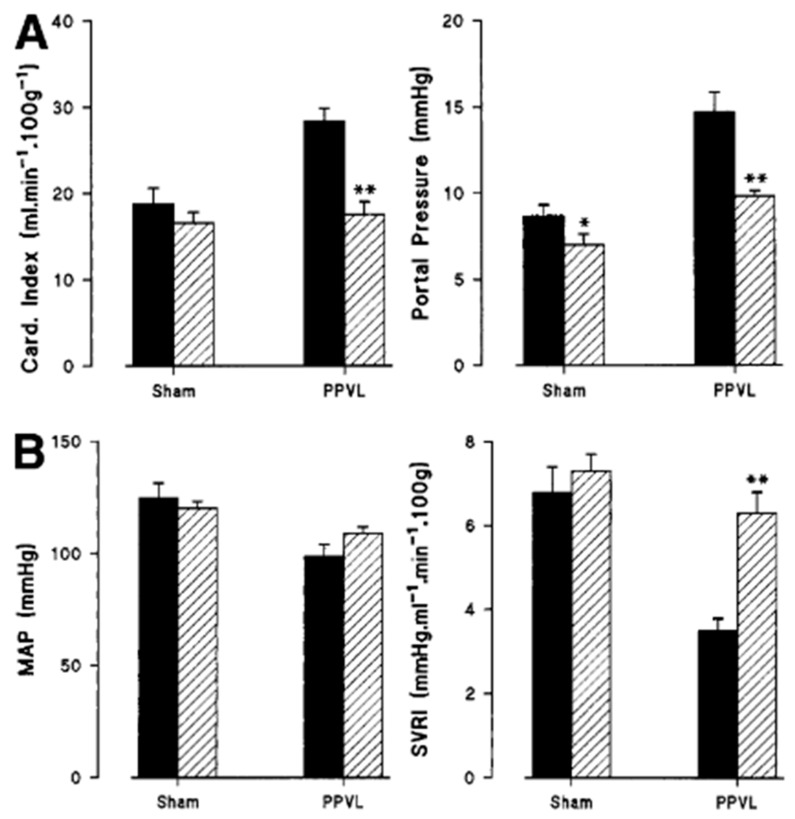
Hemodynamic studies in placebo-treated (
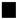
) and NAC-treated (
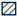
) rats. Following PPVL and administration of placebo (*n* = 6), there was an increased cardiac output, portal pressure (**A**) and a reduction of MAP and systemic vascular resistance (**B**) compared with placebo-treated sham animals (*n* = 8). These hemodynamic changes were prevented by the twice-daily administration of NAC (PPVL + NAC) (*n* = 6). Values represent means ± SEM. PPVL + NAC vs. PPVL + placebo: ** *p* < 0.0005. Sham + placebo vs. sham + NAC: * *p* < 0.05. (Reproduced from reference [38]).

**Figure 2 ijms-24-16805-f002:**
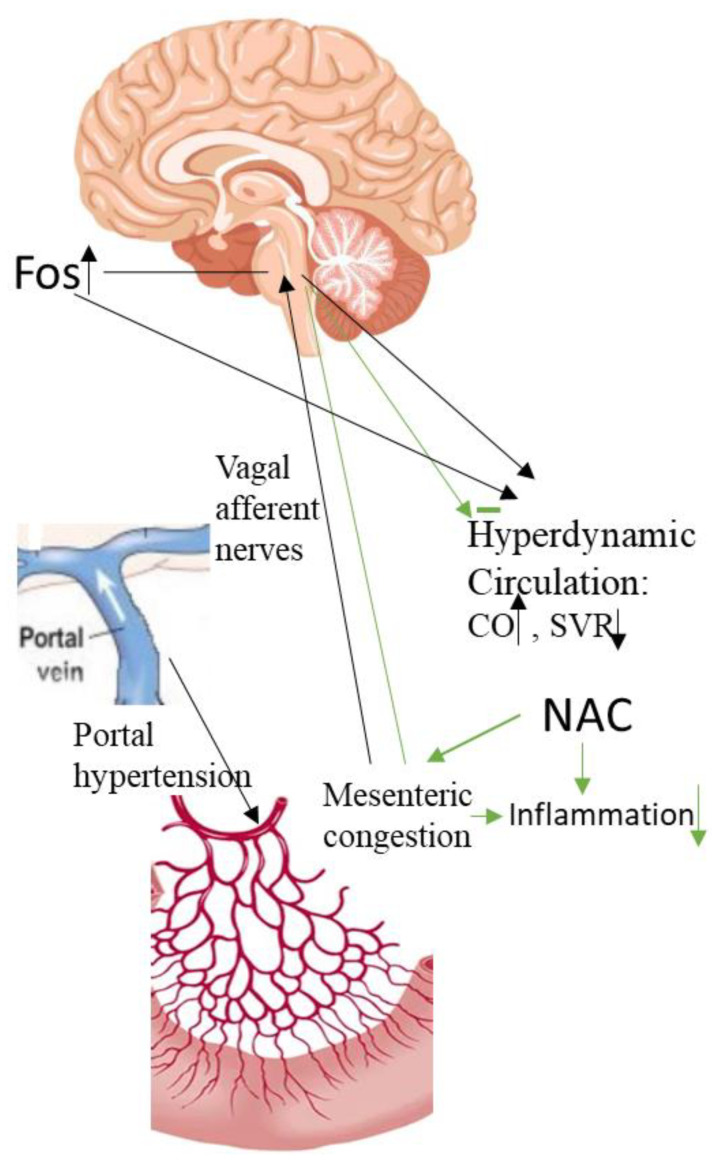
Changes in circulation indices. Schematic diagram to show the correlation among portal hypertension, mesenteric congestion, central neural activation and hyperdynamic circulation. CO: Cardiac output; MAP: mean arterial pressure; SVR: systemic vascular resistant; NAC: N-acetylcysteine. (Reproduced from reference [9]: Liu H et al., Hepatol Int 2023;17:689–697).

**Figure 3 ijms-24-16805-f003:**
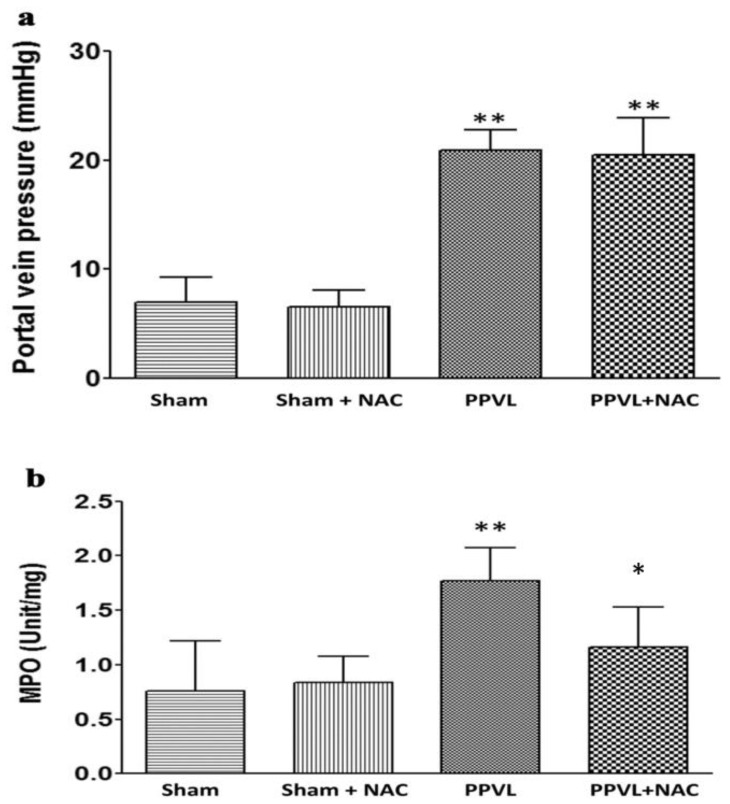
The effect of NAC on portal vein pressure and oxidative stress in mesentery area. (**a**): NAC has no effect on portal vein pressure either in Sham or BDL rats. (**b**): The effect of NAC on myeloperoxidase (MPO) status in the jejunum (** *p* < 0.01 compared with sham controls, * *p* < 0.05 compared eith PPVL group) NAC: N-acetylcysteine; PPVL: partial portal vein ligation. [Reproduced from reference [9]: Liu H et al., Hepatol Int 2023;17:689–697].

**Figure 4 ijms-24-16805-f004:**
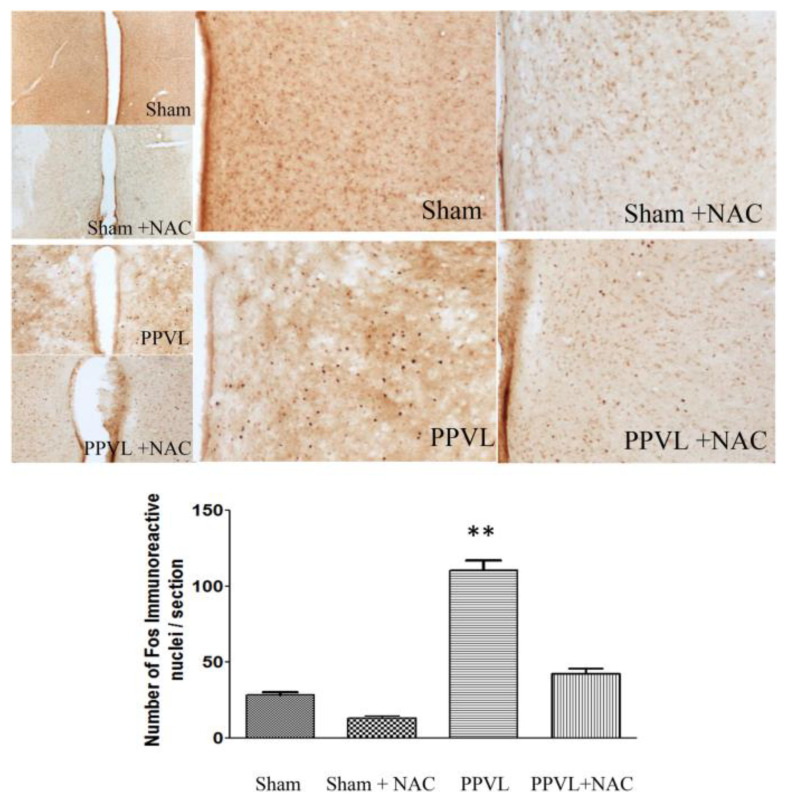
Fos density in the paraventricular nucleus (PVN) of the hypothalamus. Partial portal vein ligation (PPVL) significantly increased the density of Fos in PVN in PPVL rats. *N*-acetyl-cysteine (NAC) significantly decreased the density of Fos in PVN in PPVL rats, but did not change Fos density in PVN in sham-operated rats (** *p* < 0.01 compared with sham controls). (Reproduced from reference [9]: Liu H et al., Hepatol Int 2023; 17: 689–697).

**Figure 5 ijms-24-16805-f005:**
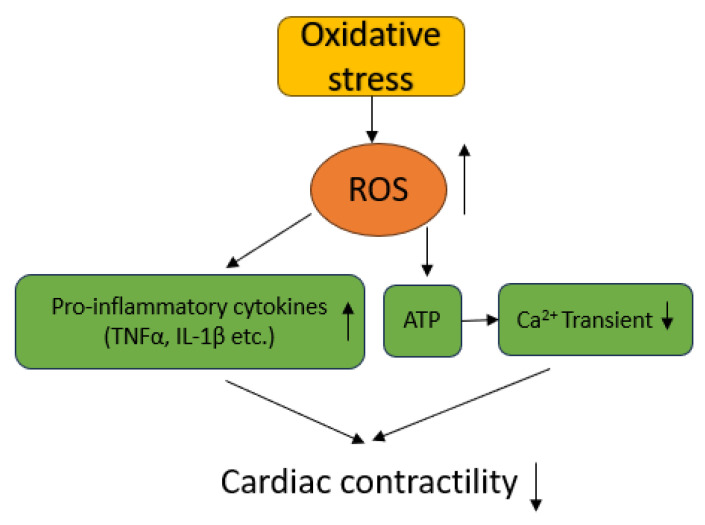
Pathogenic mechanism of oxidative stress on cardiac function.

**Table 1 ijms-24-16805-t001:** Diagnostic criteria proposed by Cirrhotic Cardiomyopathy Consortium.

Systolic Dysfunction	Advanced Diastolic Dysfunction	Areas for Future Research
Any of the following:	≥3 of the following:	Abnormal chronotropic or inotropic response
LV ejection fraction ≤ 50%	Septal e′ velocity < 7 cm/s	Electrocardiographic changes
Absolute * GLS < 18%	E/e′ ratio ≥ 15	Electromechanical uncoupling
	LAVI > 34 mL/m^2^	Myocardial mass change
	TR velocity > 2.8 m/s	Serum biomarkers
		Chamber enlargement
		CMRI

* GLS (global longitudinal strain) is reported as a negative value in echocardiography reports. Changes in GLS should be described as changes in the absolute value. LV, left ventricle; e′, early diastolic mitral annular velocity; E/e′, ratio of mitral peak velocity of early filling to early diastolic mitral annular velocity; LAVI, left atrial (LA) volume index; TR, tricuspid regurgitation; CMRI, cardiac magnetic resonance imaging.

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
