# Peer review of "Oxidative Mechanisms and Cardiovascular Abnormalities of Cirrhosis and Portal Hypertension"

_ijms, 2023, doi:10.3390/ijms242316805_

Round 1

Reviewer 1 Report

Comments and Suggestions for Authors

1. The authors should begin the article by defining portal hypertension and cirrhosis. Then they should continue with the definition of oxidative stress.

2. There are many punctuation errors in the article.

3. Once an abbreviation has been made, the word or phrase should not be written again, and vice versa.

4. There are meaningless things in some sentences. For example: 95% of energy supplied to the cardiomyocyte is derived from mitochondria (ref) (54). 

5. The authors should add a few figures for several pathogenesis, as they provided in Figure 2.

6. The conclusion section is written too briefly.

Comments on the Quality of English Language

Authors should make some changes to language and punctuation.

Author Response

  1. The authors should begin the article by defining portal hypertension and cirrhosis. Then they should continue with the definition of oxidative stress.

Reply: as per the reviewer’s comment, these terms were defined

  1. There are many punctuation errors in the article.

Reply: as per the reviewer’s comment, these were corrected.

  1. Once an abbreviation has been made, the word or phrase should not be written again, and vice versa.

Reply: as per the reviewer’s comment, the repeated definitions of the abbreviations were deleted.

  1. There are meaningless things in some sentences. For example: 95% of energy supplied to the cardiomyocyte is derived from mitochondria (ref) (54). 

Reply: since ROS are produced exclusively from mitochondria, we emphasized that the heart is enriched with mitochondria, meaning that it is vulnerable to ROS damage.

  1. The authors should add a few figures for several pathogenesis, as they provided in Figure 2.

Reply: as per the reviewer’s comment, figure 5 was added.

  1. The conclusion section is written too briefly.

Reply: more details were added to the conclusion.

Reviewer 2 Report

Comments and Suggestions for Authors

The manuscript by Liu et al. reviews the impact of portal hypertension on patients, leading to complications such as cardiovascular abnormalities. It highlights the role of reactive oxygen species (ROS) in causing intestinal issues, bacterial overgrowth, and inflammation. Portal hypertension also contributes to cardiovascular problems like hyperdynamic circulation and cirrhotic cardiomyopathy. The review highlights that oxidative stress plays a crucial role in these complications and summarizes antioxidants as potential therapeutic agents to mitigate the associated cardiovascular issues. The review is well-crafted and provides a comprehensive understanding of the mechanisms and treatment possibilities for cirrhotic patients with cardiovascular abnormalities. However, there are several suggested revisions to enhance the manuscript.

1. The review centers on the cardiovascular abnormalities observed in patients with cirrhosis and portal hypertension. While delving into the mechanisms and treatment options, it would be beneficial to include a section specifically addressing clinical manifestations and diagnostic considerations. This addition will offer clinical guidance to readers.

2. It is unnecessary to cite original figures from other studies, if so, the agreement from other authors should be stated in the current manuscript. Furthermore, note that Figure 1 exhibits blurriness, and it is recommended that the authors create their own graphs to effectively convey the review's purpose.

3. I recommend reorganizing the headings throughout the entire review. Rather than numbering them sequentially from 1 to 14, consider categorizing the manuscript based on distinct themes. For example, headings could be structured around topics such as "Mechanisms of Oxidative Stress," "Cardiovascular Abnormalities," "Treatment Options," and so forth. This approach will enhance the clarity and thematic organization of the content.

Comments on the Quality of English Language

minor editing.

Author Response

  1. The review centers on the cardiovascular abnormalities observed in patients with cirrhosis and portal hypertension. While delving into the mechanisms and treatment options, it would be beneficial to include a section specifically addressing clinical manifestations and diagnostic considerations. This addition will offer clinical guidance to readers.

Reply: as per the reviewer’s comment, the diagnostic criteria of CCM was added (table 1)

  1. It is unnecessary to cite original figures from other studies, if so, the agreement from other authors should be stated in the current manuscript. Furthermore, note that Figure 1 exhibits blurriness, and it is recommended that the authors create their own graphs to effectively convey the review's purpose.

Reply: a review paper is allowed to cite figures from original papers. The copyright permissions will be obtained. We have also sharpened figure 1, and created a new figure 5.

  1. I recommend reorganizing the headings throughout the entire review. Rather than numbering them sequentially from 1 to 14, consider categorizing the manuscript based on distinct themes. For example, headings could be structured around topics such as "Mechanisms of Oxidative Stress," "Cardiovascular Abnormalities," "Treatment Options," and so forth. This approach will enhance the clarity and thematic organization of the content.

Reply: as per the reviewer’s comment, the subtitles were revised.